# ‘The Addiction Was Making Things Harder for My Mental Health’: A Qualitative Exploration of the Views of Adults and Adolescents Accessing a Substance Misuse Treatment Service

**DOI:** 10.3390/ijerph20115967

**Published:** 2023-05-26

**Authors:** Liam Spencer, Hayley Alderson, Steph Scott, Eileen Kaner, Jonathan Ling

**Affiliations:** 1Population Health Sciences Institute, Newcastle University, Newcastle upon Tyne NE2 4AX, UKsteph.scott@newcastle.ac.uk (S.S.);; 2Faculty of Health Sciences and Wellbeing, University of Sunderland, Sunderland SR1 3SD, UK

**Keywords:** substance use, mental health, treatment, adults, adolescents, qualitative research

## Abstract

The relationship between substance use and mental health is complex, and both constitute a global public health burden. In the UK, the estimated annual financial costs of alcohol-related harm and illicit drug use are GBP 21.5 billion and GBP 10.7 billion, respectively. This issue is magnified in the North East of England, where treatment access is low and a large proportion of individuals experience socioeconomic deprivation. The present study aimed to explore the experiences of adults and adolescents accessing a substance misuse treatment service in the North East, in order to inform policy makers, commissioners, and providers of substance misuse treatment and prevention. Semi-structured qualitative interviews were conducted with an opportunistic sample of *n* = 15 adult participants (aged 18 years and over) and *n* = 10 adolescent participants (aged between 13 and 17 years). Interviews were audio-recorded, transcribed, anonymised, and analysed thematically. Five key themes were identified: (1) initiation of substance use, (2) early life experiences, (3) the bi-directional relationship of mental health and substance use, (4) cessation of substance use, and (5) accessing treatment. Future preventative interventions should focus on providing support to individuals who have been exposed to adverse childhood experiences, with treatment provision for individuals experiencing co-occurring mental health and substance use issues taking a more holistic approach.

## 1. Introduction

The relationship between substance use and mental health is complex [1], and both constitute a substantial global public health burden [2]. In 2018, it was estimated that the annual financial costs of alcohol-related harm and illicit drug use in the UK were GBP 21.5 billion and GBP 10.7 billion, respectively [3,4], and according to the Mental Health Foundation in 2022, mental health problems cost the UK economy at least GBP 117.9 billion annually [5]. Mental health disorders and problematic substance use are more common in socioeconomically deprived areas [6], and the North East of England has the second-highest low income and deprivation rates in England after inner London [7], with high levels of mental health, substance, and alcohol use problems compared to the rest of the country [8,9]. Comparison across all twelve local authorities in North East England revealed that all but one had higher-than-national-average rates of deaths from drug misuse, with the average rate across the region at 9.9 per 100,000 in 2018–2020, compared to 3.5 per 100,000 in the lowest region (London) [10]. The region has a high admission rate to hospitals for mental and behavioural disorders due to alcohol (108.8/100,000 population compared with 72.3/100,000 for England), and whilst the suicide rate in the region has remained relatively static in recent years, premature mortality from all causes for people with a mental illness remains extremely high in all local authority areas [8].

Despite high levels of alcohol harm in the North East, the proportion of adults in treatment for alcohol dependence successfully completing structured treatment (defined as individuals who did not present to treatment again within 6 months of completion), was the lowest in England (30.7%), with no significant change in trend over the last five years [11]. The proportion of individuals in the region who successfully completed drug treatment for both opiate and non-opiate dependence was also lower than the national average (71.6%), suggesting there may be significant barriers to access [12,13]. Stigmatising attitudes towards addiction in society have been linked to the marginalisation of individuals who struggle with substance use. Despite efforts to reduce stigma, the media, some drug treatment programmes, and societal norms perpetuate this discrimination. This can result in ‘self-stigma’, which creates significant barriers to treatment and recovery, as individuals internalise their substance use issues and blame themselves for the position they are in, therefore making them less likely to seek support [14].

Two forms of reinforcement have dominated addiction theory: (1) positive reinforcement, subjectively linked to drug-induced euphoria, is important for establishing drug-seeking habits and reinstating them quickly after periods of abstinence; and (2) negative reinforcement, subjectively linked to the alleviation of pain, which becomes most obvious in the late stages of sustained addiction. Whilst these theories agree about the early and late stages of addiction, they hold different views as to the point between early and late at which the diagnosis of ‘addiction’ should be invoked, the relative importance of positive and negative reinforcement leading up to this transition, and the degree to which the specifics of negative reinforcement can be generalised across the range of addictive agents [15]. Theories of development have long posited that events during sensitive periods in childhood and early adulthood can have substantial consequences on health trajectories. Adverse childhood experiences (ACEs), including physical or emotional abuse, parental substance use or mental health problems, and bereavement, are a risk factor for poor physical and mental health outcomes and substance use behaviours [16,17,18,19]. For example, Hughes et al. recently identified that adults who had reported four or more types of ACE were almost 10 times more likely to have felt suicidal or have self-harmed than those who had experienced none [20]. ACEs may also be a contributing factor to substance use in adulthood [21,22] and are associated with adolescent drug use with the potential for consequences in wider aspects of adolescent’s lives, regardless of their social, ethnic, or economic background [23]. A UK-based study which used prospective cohort data demonstrated associations between ACEs and lower educational attainment and higher risks of depression, drug use, and smoking at 16–17 years old, that remained after adjustment for family and socioeconomic factors. Further, the authors concluded that intervention strategies targeting ACE prevention and support should also target a wide range of relevant factors, including socioeconomic deprivation, parental substance use, and mental health [24].

Given the scale of substance use and mental health problems in the North East of England, reduced treatment access for both, and in acknowledgment of existing evidence which has demonstrated the relationship between deprivation and potential ACEs that can lead to mental health problems and substance use in later life, this paper builds upon a small body of existing qualitative research. We aim to provide further insight into the complex nature of this relationship, the influence of early life experiences, and how best to address the needs of this population in terms of service commissioning and delivery through exploring the experiences of adults and adolescents accessing a substance misuse treatment service in the region.

## 2. Materials and Methods

### 2.1. Study Design

As part of a broader programme of work, this study adopted a concurrent mixed methods approach through two phases of data collection, which consisted of a behavioural insights survey and qualitative interviews with adult and adolescent service users [25]. The present paper reports on the qualitative phase of data collection only. Qualitative research is recognised as enabling in-depth analysis of socially situated experiences and can help to provide insight into otherwise unknown practices, ensuring better-informed public health policy decisions through the identification of optimal opportunities for intervention, prevention, and treatment [26]. This phase of data collection utilised face-to-face semi-structured interviews with service users, to provide an in-depth understanding of the lived experiences of individuals accessing a drug and alcohol treatment service.

### 2.2. Recruitment and Sample

The study was conducted within a drug and alcohol service with two treatment centres in the North East of England. One centre focused on clinical treatment, providing specialist services including titration, stabilisation, and reduction regimes; substitute prescribing (e.g., subutex and methadone for opioid dependency); and distribution of naloxone to treat opiate overdose [27]. The other centre focused on community treatment, including the delivery of individual and group psychosocial therapeutic interventions to support recovery [28]. Participants of any age were eligible to participate in the study, as long as they were currently accessing one of the treatment centres. Over a period of nine months, one researcher (LS) spent time in both treatment centres on varying days alongside practitioners, in order to provide a varying range of adult service users the opportunity to participate in the study. Recruitment of adolescent participants (under the age of 18) was facilitated by a young person’s substance misuse worker, as these service users do not physically access the adult treatment centres, and their appointments are undertaken in schools and the community. The recruitment process was guided by the concept of ‘information power’, whereby the more relevant information a sample holds, the lower amount of participants is required [29]. Researchers are invited to position themselves and their study along a continuum, to assess an approximate number of participants needed for responsible analysis, determined by items such as study aim, sample specificity, and the strategy of analysis.

### 2.3. Data Collection

Individuals who expressed an interest in participating were provided with an information leaflet, which detailed the purpose of the study, how their data will be used, an outline of their right to confidentiality and anonymity, the voluntary nature of their participation, and their right to withdraw from the study at any time, with no impact on their care. Prior to each interview, informed consent was obtained from all participants, with parental consent and participant assent also obtained for participants under the age of 16. Participants received a GBP 10 high-street shopping voucher as payment for their time. Ethical approval was granted by the Newcastle University Faculty of Medical Sciences Ethics Committee (Ref: 1528_1/2018). A semi-structured interview schedule was developed, which was designed to guide topics of interest covered in the interviews (see Appendix A). Interviews were conducted by one researcher (LS), and were digitally audio recorded, then transcribed verbatim. The transcripts were fully anonymised, with the names of people and places omitted, and an individual identifier was allocated to each transcript. Interviews were conducted between October 2018 and May 2019.

### 2.4. Data Analysis

A reflexive thematic data analysis approach, following the guidelines set out by Braun and Clarke [30] was used to analyse the data. This approach was chosen as it is an accessible, adaptable, iterative, and theoretically flexible approach to analysing qualitative data and is particularly useful in the context of applied health research, which often involves complex and multi-layered issues and aims to inform policy and practice [31]. All data were coded and organised using NVivo for Mac release 1.7.1 (QSR International; Burlington, MA, USA). Initial data coding was undertaken iteratively by LS, using a combination of inductive approaches, guided by patterns, themes, and categories that emerged from the data, and deductive approaches, guided by existing literature. Whilst the interview schedule was originally designed to explore participants’ experiences of access the treatment service, emergent themes were discussed and developed with HA. A dominant theme that participants self-reported was their mental health symptomology; interestingly, mental health status was not a specified inclusion criterium and their participation in the study was not dependent on a formal mental health diagnosis. The original codes and sub-theme categories were classified into five main themes that were considered significant areas of relevance to policy makers, commissioners, and providers of interventions that aim to prevent and treat co-occurring substance use and mental health disorders. Verbatim participant quotations from interviews are used to illustrate the themes.

## 3. Results

### 3.1. Sample

The adult participants (*n* = 15) were all aged 18 years old and over (specific age not collected); 11 participants were male and 4 were female. Adolescents who participated (*n* = 10) were aged between 13 and 17 years old (median = 14 years); nine participants were male and one was female, which is representative of the gender balance of participants accessing treatment nationally [13,32]. All participants identified as white British, which remains largely representative of demographics across the region in question (North East England) [33]. Minimal demographic information was collected about participants at the request of the treatment service. We did not wish to deter individuals from participating through collecting and reporting potentially identifiable information. Interviews lasted between 10 and 50 minutes (mean = 30 min). Five key themes were identified during analysis: (1) initiation of substance use, (2) early life experiences, (3) the bi-directional relationship of mental health and substance use, (4) cessation of substance use, and (5) accessing treatment. 

### 3.2. Initiation of Substance Use

Participants’ journeys into using both licit and illicit substances were inherently personal, and as such, varied greatly. However, patterns of initiation did emerge, with similar recollections being shared across participants of all ages. Adult and adolescent participants made specific reference to their perceived self-management of emerging negative mental health symptoms as a motivating factor in the initiation of substance use. Participants described using a range of substances (licit and illicit), as well as the misuse of prescription drugs, as a way to cope with or ‘blank-out’ symptoms associated with poor mental health, including low mood, depression, stress, and anxiety. This was often as a consequence of a build-up of ‘trauma’ or following a particularly turbulent event in their lives such as the breakdown of a relationship, and turning to substances was perceived by participants to be a reasonable coping mechanism and a way to improve their situation.


*“I had a problem with tramadol. I was self-medicating. Well, I was self-medicating on more than just tramadol, to be fair. It was anything I could get my hands on. At first, I thought it was just helping with my mental health.”*
(Adult service user 3, male)


*“It got to a point where every time I’d have a drink, I’d have a smoke [cannabis]. I just felt like I could relax, and not have to think about anything.”*
(Adolescent service user 8, 14-year-old male)

In addition to the self-management of negative mental health syptoms, a dominant sub-theme of initiation amongst both adult and adolescent participants arose. It was described that their initial exposure to substances, and the initiation of their substance use was often strongly linked to socialising, with friends, acquaintances, and social groups being identified as a common gateway. Many participants articulated a desire to ‘fit in’ and felt that substances acted as a ‘social lubricant’, especially in contexts that required them to engage with new acquaintances who already used these substances, suggesting that in some circumstances the influence of peers can lead to initiation. However, some participants also referred to engaging in substance use simply due to boredom, and curiosity about the effects on themselves, particularly when at a younger age.


*“They weren’t friends, they were just using me for entertainment, but I thought they were friends. I used alcohol as a tool to cling onto people.”*
(Adult service user 1, male)


*“I was at one of those carnivals. People were having it [cannabis], and then, I was with them, and they said, “Do you want to try it?” Then, I tried it and then glued on.”*
(Adolescent service user 3, 14-year-old male)

In addition, several adult participants also spoke about initiation of substance use in the context of their working lives, and how the environment in which they worked offered opportunities for them to engage in substance use. In some cases, this was through increased socialisation, for example drinking alcohol with work colleagues; however, this would occasionally act as a gateway to other substances, such as cannabis and ‘party drugs’, such as stimulants including cocaine and amphetamines, and hallucinogens such as methylenedioxymethamphetamine (MDMA), also known colloquially as ecstasy.


*“I was in and around drugs and drinking with my job. I started off smoking a bit of weed and just went from that to the party drugs.”*
(Adult service user 9, male)


*“It was fun when I was at college. Working in various bars and that. That’s a hard one, because it was always beer o’clock whenever your shift finished.”*
(Adult service user 8, male)

One participant also referred specifically to engaging in a particular amphetamine-type stimulant (speed) in their workplace context, and whilst this was initially perceived as having beneficial effects on their capability at work, this contributed to a cycle of drug addiction.


*“She [colleague] came in one day really buzzing about how this boyfriend of hers had injected her with speed. I thought, “Oh, I’ll have a go on that.” and that started my first proper drug addiction.”*
(Adult service user 5, male)

### 3.3. Early Life Experiences

A particularly strong theme which emerged amongst most adult participants was frequent and detailed references to significant events experienced earlier in their lives, which had a lasting impact. Many participants shared many stories from their childhoods and early adulthoods, which encompassed experiences of poverty, adversity, and trauma, including physical, sexual, and emotional abuse. These early experiences were viewed as playing a big part in shaping participants’ futures. Through self-reflection and navigating their pasts in retrospect, participants were able, in some cases through having engaged in talking therapies, to better understand and reconcile their subsequent substance use behaviours in the context of adverse and traumatic early experiences.


*“I had a very rough childhood. I was in and out of foster homes and children’s homes. I was abused physically, sexually, mentally, and emotionally from the age of five.”*
(Adult service user 4, female)


*“Looking back on it, a really bad childhood. There was abuse, there was alcohol involved and stuff like that. I got through it [‘bad childhood’] just by withdrawing from it.”*
(Adult service user 5, male)

Several adult participants also spoke about their exposure to substance use, particularly alcohol consumption, in early life, and often with close family members, such as a parents and caregivers. These early recollections included being ‘forced’ to consume alcohol, and witnessing abuse perpetrated by family members who were under the influence of alcohol against other family members. Being witness to alcohol use at an early stage in participants’ lives may have affected their relationship with substances in later life through altering their understanding and perception of what ‘normality’ was in the context of substance use.


*“I started drinking when I was a child. I was forced to drink, by my stepfather. He would just get whisky out and just put it into a cup and just force you to drink it.”*
(Adult service user 4, female)


*“My dad was a big drinker and he’s not a nice guy. He mistreated my mum and blah, blah, blah. My mum’s dad, he was an alcoholic as well and he used to mistreat my grandma.”*
(Adult service user 9, male)

Many adolescents we spoke to shared similar stories of adversity and trauma, including parental relationship breakdowns and parental mental health issues, compared to adult participants. In many cases, these participants were still living through these situations and were in the process of navigating challenging, turbulent, and often disruptive family circumstances. As their experiences were still so recent and ‘raw’, adolescent participants were understandably not able to be as objective regarding these experiences. However, comparisons can be drawn between the experiences they were actively living through and the narratives shared by adult participants reflecting back upon the early experiences that contributed to their substance use journeys.


*“My mum and dad got divorced, and then I lived with two out of three sisters, then my other sister got moved into my other sister’s house. Now, it’s just me, my sister, and my mum.”*
(Adolescent service user 3, 14-year-old male)


*“My dad suffered from huge bouts of depression. His sister overdosed and he didn’t really want to make that mistake, but he decided to [use substances] when he had me and my sister. My mum left him, and his life just went downhill.”*
(Adolescent service user 10, 16-year-old male)

### 3.4. The Bi-Directional Relationship of Mental Health and Substance Use

Many adult participants referred to episodes of poor mental health such as periods of depression during adulthood, which led to associated increases in their substance use. These episodes were often triggered by significant and/or traumatic life events. Participants referred to continued self-management of mental health issues through increased self-medication with substances, which several participants had stated was the original initiation point for their substance use careers. However, participants also reflected that as time progressed, with both their substance use and mental health issues spiralling, they were unable to establish if they were self-medicating to improve their mental health, or if this substance use was in fact contributing to a worsening in their mental health symptoms.


*“I did reduce once, which I stopped for three months. Then my dad died suddenly, and I went out of control. Mental health [services] was involved.”*
(Adult service user 10, male)


*“I had a massive bout of depression, and I was placed on the sick by my then employer, I relapsed, and had a spell in hospital for a detox.”*
(Adult service user 11, male)

One participant spoke specifically about their inability to engage in physical exercise due to a back injury. Exercise had previously been their primary coping method when they felt their mental health declining, and this barrier led to them engaging in increased alcohol use to self-manage their mental health, which unfortunately continued to deteriorate.


*“I generally get through things, by running and exercise, for my mental health. I couldn’t do that, so I went on a bit of a downward spiral, I was drinking quite a bit.”*
(Adult service user 15, female)

Similar to experiences they shared regarding the initiation of their substance use, several adolescents also spoke about their continued use as a method of maintaining or even improving their mental health. Whilst these participants did acknowledge a desire to cease using substances, they felt that they were unable to, as they perceived the effects on their mental health to be worthwhile, making particular reference to the ability it gave them to ‘forget’ about their issues.


*“I just feel the need to have a joint […] it just takes my mind off everything, it’s like soon as you have a joint you don’t think about nowt.”*
(Adolescent service user 1, 16-year-old male)


*“I feel like, at the minute, I can’t [stop] because I just like that feeling. When I’m stressed, I like the feeling of being stoned, because you just forget everything.”*
(Adolescent service user 3, 14-year-old male)

Some adult participants acknowledged the notion of them masking or covering up mental health issues with their substance use, rather than directly admitting they required mental health support. They implied that being perceived as being under the influence of substances, having issues with substances, or even accessing treatment for problems with substance use was preferable socially to being perceived as having mental health issues or even acknowledging personally the presence of a potential mental health issue. One participant made explicit reference to the perceived stigma associated with mental health issues.


*“In the past I’ve tried to cover [mental health issues] up because there’s a lot of stigma, instead it’s, ‘oh, he’s had too much drink’.”*
(Adult service user 8, male)


*“I’m just trying to fix the other problems, mentally, that have gone on in my life. There was a big cloud, masked over by just flying around the world and getting drugged up and pissed up.”*
(Adult service user 9, male)

Several participants went on to discuss reaching a point of ‘acceptance’ that their substance use was in fact contributing to their worsening mental health issues, rather than being instrumental in alleviating symptoms and supporting them to feel better able to cope, as they previously felt during periods of self-medication. In some participants’ circumstances, this acceptance was reached when they identified that the levels at which they were consuming substances had substantially increased, without the previously perceived improvement in symptoms.


*“My mental health and my drinking kind of interconnect. When my mental health goes bad my drinking increases, when my drinking increases my mental health deteriorates.”*
(Adult service user 4, female)


*“The addiction was making things harder for my mental health. Then when I started doing some of the classes and realised that it’s a massive circle of mental health as well as drug addiction.”*
(Adult service user 3, male)

Some adolescents appeared to have a well-developed understanding of the bi-directional nature of the relationship between mental health and problematic substance use and were forthcoming about their mental health issues. This acknowledgement and acceptance may be a consequence of the early and preventative nature of interventions provided to the adolescents by substance misuse workers, which aimed to provide education and increase awareness in the context of the adolescent’s personal circumstances.


*“They [drugs] send you fucking barmy like, proper barmy. I was getting locked up every week, about once or twice a week or something. I’d be in jail now if I was still on them.”*
(Adolescent service user 6, 17-year-old male)


*“The worst thing is I’m starting to realise every time I drink, the more suicidal things become. It has become a steak knife to my wrist, deeper to my neck, to my stomach.”*
(Adolescent service user 10, 16-year-old male)

### 3.5. Cessation of Substance Use

Many of our adult participants shared experiences of serious and sometimes life-threatening physical health issues caused directly by their substance use. Up until this point, many participants had been oblivious to, or chose to ignore, the detrimental effect their substance use was having upon their physical health. These events were perceived by participants to be key turning points for them. A facilitating factor in the decision to engage in treatment, these experiences helped participants realise that they needed to make substantial changes in their lives with regard to their substance use behaviours, or else they risked causing themselves greater physical harm.


*“I nearly died because of [heroin]. I had fluid on the lungs, I was rushed into hospital, and the doctor said that if I didn’t come, I would have died.”*
(Adult service user 2, male)


*“Over the last couple of weeks my health declined so much, to the point that my liver and my kidneys were getting damaged.”*
(Adult service user 3, male)

Whilst most adolescents we spoke to did not report having experienced serious negative impacts upon their physical health, the potential of this was something they were aware of, having engaged in therapeutic relationships with substance misuse workers. Learning about the potential dangers of substance use on their physical and mental health, along with witnessing older individuals (often family members) further along their substance use careers and experiencing difficulties, were strong motivating factors in their desire to cease substance use, or at least encouraged a desire to reduce their consumption moving forward.


*“When you see how bad it could affect you if you’re stoned, and just the little things like what it can do to you and stuff. It was putting me off.”*
(Adolescent service user 3, 14-year-old male)


*“Weed, like if you’re young and you smoke it like, it makes your brain not function properly. Drinking like doesn’t just affect your brain, it’s like all your insides and stuff.”*
(Adolescent service user 7, 14-year-old male)

The majority of adult participants expressed a desire for change and wanted a ‘better life’—often after many years of being stuck in the same cycle of substance use. Examples of this included wishing to sustain meaningful employment, establishing a stable home, and starting a family. These were also important motivating factors for participants who had yet to reach a point where their substance use had manifested serious physical health problems but were aware of the potential for this to occur if their behaviours continued in a similar pattern.


*“I’m trying to change my life around. I don’t want to be on medication anymore. I’ve got myself into employment which I’ve been promised will become permanent.”*
(Adult service user 7, male)


*“To be drug-free, get my life back on track, get my own place, get it furnished, and eventually get a job. Meet a woman and start my own family, basically, like anybody else.”*
(Adult service user 2, male)

Some adult participants expanded on this, reflecting on how their substance use had caused damage to previously established relationships with family members, and they expressed a desire to try to undo some of this damage, as well as prevent further damage from occurring. Participants who were parents spoke about how important it was for them to be able to stop using substances to prevent losing the custody of their children, or to establish a stable period in recovery in order to regain custody of, or contact with, children already removed.


*“My little lad came for a cuddle, and I had a glass of wine in my hand. He knocked my glass, and it went flying. I was fuming with him, just like, “Stupid fucking thing.” I realised, at that point, giving my son a cuddle was not my first priority.”*
(Adult service user 4, female)


*“This time, I want everything gone and just to the point where I go out with my family and, like my kids and stuff.”*
(Adult service user 14, male)

This desire for change seemed to be a particularly strong important motivating factor for adolescents, who were at a crucial point for change in their lives regarding making decisions about future education and career possibilities. Adolescents spoke about wanting to go on and fulfil career ambitions, which they believed would be restricted if their substance use continued or escalated. Adolescents also spoke about wanting to improve their relationships with family members, which had been damaged as a consequence of their substance use.


*“I want to be something that’s handy because my fun is, like, mucky fun. I like going and getting muddy and stuff like that. So, I think I either want to be a joiner, a plumber, electrician, a bricklayer, or a scaffolder.”*
(Adolescent service user 4, 13-year-old male)


*“If I got caught with something like weed, got anything on my record it wouldn’t get me a good job and stuff like that.”*
(Adolescent service user 7, 14-year-old male)

### 3.6. Accessing Treatment

Several adult participants spoke positively about the impact that substance use treatment had on their lives, which is unsurprising as the sample consisted of participants who were engaged in treatment. The compassionate and supportive nature of the substance misuse workers they were engaged with was incredibly important to participants. The fact that they did not feel judged by workers, or by other service users, made it much easier for participants to engage with the treatment options offered to them, and encouraged them to remain in treatment.


*“They were very supportive, empathetic. They could see I meant it [wanting to stop using] and I was desperate […] I was relieved I’d been forced into making that step.”*
(Adult service user 4, male)


*“It was cathartic, none of them ever judged me, not the service users or the staff, which is a wonderful feeling […] I was given a chance and I grabbed it.”*
(Adult service user 5, male)

One participant expanded upon this, sharing that he felt he had engaged more positively with treatment than he anticipated. He felt that he had learned a lot about himself, and experienced surprising levels of self-growth during the process.


*“I’ve learned so much about myself and I’ve gone quicker with it than I thought I would. I’ve grown as a person; it’s been a massive eye-opener.”*
(Adult service user 9, male)

Overwhelmingly, despite some indication of initial concern and reluctance to engage with services and with treatment, adolescent participants also spoke positively about the therapeutic relationships that they had formed with their substance misuse workers. The nature of these relationships was key to ensuring that adolescents remained motivated to engage in treatment, with participants speaking highly about the support they received from workers, and the benefits they gained from accessing this support.


*“I thought it was going to involve social workers, and I didn’t want to go. Then I went and found out what it was like and thought, ‘I’ll come back’.”*
(Adolescent service user 1, 16-year-old male)


*“It warms my heart when I’m with them because I know I’m supported. I never thought I would feel that […]. When I cannot be arsed [to go to the treatment service], I force myself because I know it’s beneficial.”*
(Adolescent service user 10, 16-year-old male)

Adolescent’s substance use behaviours were also influenced by the educational interventions delivered to them. Through this approach, adolescents were informed about the range of substances they may be exposed to, and the potential deleterious impacts that these could have on their physical and mental health. This provided adolescents with accurate information, to help them make their own informed decisions about whether to use substances.


*“We take stuff back with us, they don’t just give you daft fucking information; they’ll tell you proper stuff.”*
(Adolescent service user 6, 17-year-old male)


*“The fact that it’s getting me off the drug, and it’s someone to talk to about drugs as well, isn’t it? It’s just a help.”*
(Adolescent service user 3, 14-year-old male)

Several adult participants spoke about potential barriers to accessing treatment. These individuals felt there was a perceived stigma around mental health, substance use, and accessing treatment. This was further compounded by their own personal feelings of shame and embarrassment, with most participants initially struggling to accept that they had a ‘problem’. Participants felt that this stigma was experienced at different levels, dependent on the type of substance they were seeking treatment for—with issues relating to illicit substances (particularly Class A drugs) perceived as more stigmatised.


*“People who have alcohol issues don’t get tarred with the same stigma as people who have issues with heroin and cocaine. There’s a definite divide between opinions of the two.”*
(Adult service user 4, female)


*“There’s a stigma about this place. Obviously, these are great with me here, don’t get me wrong, I’m lucky.”*
(Adult service user 12, male)

Several adult participants suggested that perceptions of shame and stigma can prevent individuals from being motivated, emotionally prepared, or willing to engage with treatment, with the notion of ‘readiness’ being important for treatment to be successful. Different service options and treatment pathways were available, and therefore it was often a case of trial and error to find an appropriately tailored treatment plan for individuals, as there is not a one-size-fits-all approach that works for all individuals.


*“It is trial and error, and you can’t be too disheartened if something doesn’t work for you, for whatever reason.”*
(Adult service user 9, male)


*“Some people aren’t ready. Some people suffer from anxiety big time, and [attending] a group of people, it’s scary. I mean, I was like that myself.”*
(Adult service user 10, male)

Some participants highlighted a particular limitation of the treatment services they access, referring to the lack of provision available out of hours, especially on evenings and weekends. Participants acknowledged that the treatment service they accessed, as well as other services across the country, are limited by the scope of their funding and staff availability. However, participants perceived that on evenings and weekends, there is often more temptation for people to engage in using substances and a greater risk of having a mental health crisis or emergency, and that support should be made available then.


*“When do people struggle most? On a weekend. There is nothing [no services operational] there. I know you can’t babysit people 24/7, and there’s a problem with funding.”*
(Adult service user 4, male)


*“Late at night, there is no one. It’s nothing to do with [the service], because they can only run what they’re commissioned to run.”*
(Adult service user 11, male)

## 4. Discussion

The objective of this paper was to explore the experiences of adults and adolescents accessing substance use treatment in the North East of England, with a particular focus on the complex relationship between participants’ substance use and mental health, and their early life experiences, in order to inform treatment service commissioners and providers of how best to address the needs of this population. Participants recounted personal and nuanced accounts of their journeys through substance use. Commonalities between participants’ experiences of initiating substance use were identified, and similar patterns of adverse childhood experiences were evident across the sample, as was a description of the bi-directional nature of the relationship between substance use and mental health. Motivating factors for the cessation of substance use and engagement with treatment services were identified, as were strengths, limitations, and barriers to accessing treatment.

As established in previous research, substance use initiation can occur through a variety of means [1], with individuals often starting to use substances as a form of ‘self-management’ to provide temporary relief from symptoms of poor mental health, such as anxiety and depression [2,34,35,36]. In addition to the influence of negative mental health symptomology, initiation can also occur through exposure in friendship groups, with individuals being influenced by the attitudes and behaviours of peers who use licit and illicit substances in social situations [37,38,39]. Whilst individuals may inherit traits which predispose them to engaging in risky behaviours such as substance use or developing mental health issues [40,41], it is perhaps more important to recognise the contribution of environmental factors, including exposure to parental substance use [42]. In the present study, both adolescents and adults shared experiences of adverse childhood experiences (ACEs), which may be an issue that current service provision does not adequately cater for. Prevention and early intervention efforts should focus on providing support to individuals who have been exposed to ACEs, which may involve the development of child-focussed strategies for children and adolescents [43] and trauma-informed approaches that recognise the impact of trauma on individuals’ lives, aim to create safe and supportive environments for healing and recovery, and are designed to reduce the likelihood of the development of problematic substance use and/or mental health issues [44,45,46].

Individuals may use substances to cope with symptoms of poor mental health, which can lead to a perceived improvement in mental health [2,34,35,36]. However, continued substance use can exacerbate existing mental health issues, mask underlying issues, and contribute to the development of new mental health concerns [47,48]. These findings demonstrate the influence of both positive and negative reinforcement in the initation and continuation of substance use [15]. Masking of issues can also occur as a consequence of perceived stigma, or ‘self-stigma’, around accessing treatment for mental health issues, which can have a significant impact on individuals’ likelihood to engage with treatment due to fear of judgment or discrimination and delay or prevent them from receiving necessary support [14,49]. There is already existing evidence that this stigma is experienced more greatly by females [50], which may explain why the majority of individuals accessing substance misuse treatment are male. Treatment provision should take a more holistic approach and address mental health and substance use issues co-currently when necessary [51,52]. However, despite a push for ‘no wrong door’ policies, which aim to ensure that individuals with co-occurring issues can receive appropriate care and support in a timely and effective manner, regardless of where they enter the system [53], mental health services are often unable to provide support until substance use issues have been addressed [54].

The cessation of substance use can be a complex process, and success can be influenced by a range of factors, both for individuals accessing treatment and recovery in cases without formal treatment or support. Declining mental health or a reduction in the perceived positive effects of substance use on individuals’ mental health and a decline in physical health can act as key motivating factors, particularly those who have long-term substance use careers [55]. Individuals can also ‘mature out’ of the excessive use they engaged in during early adulthood, whereby involvement in new roles and activities such as starting a family or building a career conflicts with these substance use behaviours [56]. This desire for an improved life, which can also include pursuing educational or employment opportunities or starting a family, can motivate individuals to make positive changes in their lives and seek treatment for substance use disorders [57]. This is particularly salient for adolescents, who have the potential to fulfil career-related ambitions [58,59], and it is therefore important to acknowledge that adolescents may have different needs and requirements in relation to treatment access.

The strength of educational interventions provided to adolescents helps to secure buy-in from treatment-seeking individuals. These interventions can provide individuals with valuable information about substances, addiction, and the potential risks of continued use, and this can support individuals to make informed decisions and develop effective coping strategies [58,59]. The relationships formed between clients and substance misuse workers is another key factor in success, and individuals who feel a strong connection with their workers are more likely to engage with treatment and achieve positive outcomes [60], with trust a key component of effective therapeutic responses [61]. A reported limitation of many drug and alcohol treatment services is the lack of out-of-hours provision, particularly in the evenings and weekends [62]. This can be challenging for individuals who need support outside of regular business hours, and the unavailability of this support can increase the risk of relapse, as well as lead to additional pressures on primary care, through increased emergency hospital presentations for overdoses and mental health crises [63].

### 4.1. Strengths and Limitations

A strength of this work is the qualitative methodology employed, which provides rich insight into the contextual factors that surround a group of participants in an area of low income, high deprivation, and high rates of mental health disorders and problematic substance use, responding to a growing impetus to include the views and experiences of those whose voices are often overlooked or under-represented in this area of research [64]. Of particular originality, this study included both adult and adolescent service users, which allowed for contrasts to be drawn between these population groups. However, limited demographic information was collected about participants at the request of the treatment service. This meant that it was not possible to clearly distinguish findings between possible sub-groups of participants and, for example, make comparisons between those with and without formal mental health diagnoses or between those using different substances. Recruitment was only undertaken in one treatment service in one region of England, and as the participants were engaged in treatment, the views of individuals who did not engage or withdrew from treatment are not represented. Whilst the total sample size was relatively small, sample size and recruitment were guided by the ‘information power’ the sample held, where the more relevant information a sample holds, the lower amount of participants is required [29]. Conducting a small number of interviews can enable researchers to become more closely associated with the data through a considered approach to the context and complexity of individuals’ experiences, thus strengthening the rigour of in-depth qualitative analysis [65,66,67]. The sample did not have an equal gender balance (with 80% of participants identifying as male), and this is representative of the unequal gender balance of both adults and adolescents accessing drug and alcohol use treatment in the region and nationally [11,12,13,32].

### 4.2. Implications

National policymakers should highlight that health issues inclusive of substance misuse and mental health are more prevalent in deprived areas such as the area included in this study. Therefore, the development of future drug and alcohol prevention and treatment programmes should carefully consider the influence of ACEs, both in the context of delivery to adolescents (and the situations and contexts they presently inhabit) and adults (the long-lasting influence of these experiences). Treatment programmes and resources should be targeted at these areas accordingly. The early identification of issues associated with both substance misuse and mental health is important. Preventative work in schools could educate young people about the relationship between substance misuse and mental health, highlighting the potential for one issue to perpetuate the other. Due to the complex and bi-directional nature of the relationship between mental health and substance use, these findings highlight the need for interventions to take a more joined up, integrated and holistic approach [68] to address substance use and mental health concerns together. Future research could build upon the findings presented here, and it would be advantageous for research to be conducted with different population groups, including individuals who declined to engage or withdrew from treatment, and in more varied geographical and socioeconomic contexts. A novel approach such as Participatory Action Research (PAR) could be employed in order to encourage greater involvement of service users in service design [69]. Through this approach, both qualitative and quantitative research methods could be employed to explore, with service user populations, how services are developed and perceived and whether the impacts are positive. Larger-scale research such as this may allow for more nuanced recommendations to be made to different agencies, such as schools, early intervention services, and drug and alcohol treatment services.

## 5. Conclusions

This paper explored the experiences of adults and adolescents accessing substance use treatment, with a particular focus on the relationship between participants’ substance use, mental health, and early life experiences, in order to better inform how the needs of this population are addressed in terms of service commissioning and delivery. Preventative efforts should focus on providing support to adolescents and adults who may have been exposed to ACEs, with treatment provision for individuals experiencing co-occurring mental health and substance use issues taking a more holistic approach. Efforts should be made to reduce stigma associated with accessing treatment for both mental health and substance use issues, and greater involvement of service users in service design should be facilitated.

## Data Availability

Summaries of the qualitative data presented in the study are available upon reasonable request from the corresponding author. These data are not publicly available due to containing potentially identifiable information about participants.

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
