# Peer review of "‘The Addiction Was Making Things Harder for My Mental Health’: A Qualitative Exploration of the Views of Adults and Adolescents Accessing a Substance Misuse Treatment Service"

_ijerph, 2023, doi:10.3390/ijerph20115967_

Round 1

Reviewer 1 Report

The submitted manuscript entitled "‘The addiction was making things harder for my mental health’: A qualitative exploration of the views of adults and adolescents accessing a substance misuse treatment service" presents a report from the qualitative study among group of addicted individuals. As the Authors claim, this kins of analysis is important in order to obtain in-depth insight into processes associated with development of the addiction and factors connected with treatment. The results are in accordance with contemporary knowledge about the addictions and its treatment. However, I find the methodology used beneficial. After reading the manuscript I have some suggestions which could improve the text.

#1. I think that the participants should be beter described in terms of their age, economic status, co-occurance of other mental health problems etc.

#2. How the sample size was determined. Which criteria the Authors used to end the procedure of recruitment to the study.

#3. If the inteview was semi-structure, I think that future studies could benefit from adding a structure of the interview as an appendix.

#4. According to bidirectional associations between mental health problems and addiction, the Authors could divide their suggestions into several parts and present it as suggestions for institutions, parents, schools, etc. 

#5. I have some dobts about an interview which lasted for 10 minutes. Which were the reasons of longer or shorter duration of the interview.

#6. I am wondering whether some quantitative analysis could be also used in order to show how frequent some themes appeared in the interviews. 

#7. In the introduction, some theiries of addiction development could be mentioned (https://www.nature.com/articles/npp2013261; 10.1177/0193841X07307315)

Author Response

Please find attached response.

Reviewer 2 Report

Dear authors,

Kindly find the following suggestions:

There was no information on the participation's status of mental health. This might be important to select participants with mental disorders for this study.

The authors did not clarify the member checking or participation validation.

It will be clearer if the numbers of adults and adolescents were indicated for example in line 161 which is related to mental health.  

Some of the statements under the theme "initiation of substance use" might also be considered to be part of mental health or not.

Thank you.

Author Response

Please find attached response.

Reviewer 3 Report

This is a well written description of narrative stories from adult and adolescent substance users, with a particular focus on adverse childhood experiences. This is interesting and of value, and the main concern is that much of what is reported is already well established from other studies. Thus, it is unclear what is new in this manuscript. Thus, the paper would be much improved by making much clearer comparisons to what is known, and more importantly, what is unclear, in the previous literature. This would help bring out what this manuscript offers that is new.

Author Response

Please find attached response.

Round 2

Reviewer 1 Report

Thanks to the Authors for their revision. The current form of the manuscript is improved compared to the previous version. I would like only to stress that some reflection on the criteria of information power should be included when regarding the recriutiment method. Moreoover I think that one additional sentence on the limitation connected with lack of possibility to clearly disntugiushed between possible subgroups of participants shoul be included in the discussion.

Author Response

Dear reviewer,

Many thanks for taking the time to review our transcript again, it is much appreciated. We have made changes at '2.2 recruitment and sample' and '4.1 strengths and limitations’ (highlighted in yellow), in response to your additional comments. 

Kind regards,

Authors

Reviewer 3 Report

now acceptable

Author Response

Dear reviewer,

Many thanks for your positive response. We are very grateful for your time reviewing our manuscript.

Regards,

Authors